# MULTI-VISION MULTI-PROMPT FOR FEW-SHOT LEARNING IN VISION-LANGUAGE MODEL

## ABSTRACT

In vision-language models such as the Contrastive Language-Image Pre-Training model (CLIP), prompt learning can efficiently and rapidly adapt to specific tasks in few-shot learning. Previous methods for prompt learning often rely on a single prompt. However, a single prompt may not accurately distinguish between different categories, especially when a category has multiple features and contextual connections in a few-shot learning environment. While the performance of few-shot learning can improve through meta-learning or image augmentation strategies, these approaches may increase computational cost and affect accuracy. To address these issues, we propose a new method called Multi-Vision Multi-Prompt (MVMP), designed for CLIP in a few-shot learning environment. Instead of increasing the number of model parameters, MVMP employs multiple prompts at different stages of the training process and averages the predictions. Additionally, we present a mixed self-augmentation framework and text distillation to further enhance the model's performance. Extensive experimental validation demonstrates that our approach significantly outperforms the state-of-the-art in the few-shot learning classification tasks, improving accuracy by 4.6% and 2%.

## 1 INTRODUCTION

Vision-language models like the Contrastive Language-Image Pre-Training model (CLIP) (Radford et al., 2021) have demonstrated remarkable adaptation capabilities across various classification tasks, especially for few-shot learning (Sung et al., 2018). By utilizing an adversarial loss to align image-text pairs better, these models have demonstrated their versatility for specific downstream tasks. However, the large number of parameters in these models poses a challenge for quick and efficient adaptation to specific tasks (Zhou et al., 2022b). Therefore, ongoing research aims to maintain their strong adaptive capacity without increasing complexity and also enhance adaptive efficiency.

One common solution is the introduction of prompts for adaptation. Many existing methods utilize learnable prompts to efficiently adapt to few-shot learning tasks (Zhou et al., 2022a; Khattak et al., 2023; Zhou et al., 2022b). However, these methods often rely on a single prompt, which may not be sufficient to distinguish between the various categories in the dataset effectively, thus impacting the accuracy of the model (Chen et al., 2023). In response to this problem, PLOT (Chen et al., 2023), as a pioneering method, uses multiple prompts through the application of optimal transport (Peyré & Cuturi, 2019). This method has resulted in significant improvements across several datasets compared with a single prompt, such as the ImageNet dataset (Deng et al., 2009), with a 3% enhancement in performance. However, incorporating optimal transport in PLOT increases the optimization complexity, posing a higher computational burden on CLIP.

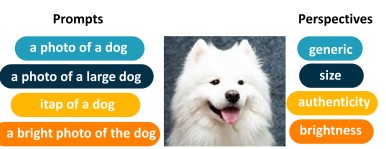

Figure 1: An example of a dog image examined through various lenses using multiple prompts, each with different perspectives.

Additionally, although there are certain advanced optimization tools for few-shot learning, such as meta-learning (Ni et al., 2021; Sun et al., 2019) and image augmentation (Osahor & Nasrabadi, 2022; Ni et al., 2021), these strategies may not always be applicable to large-scale vision-language models. Because these approaches typically introduce additional parameters or employ methods like

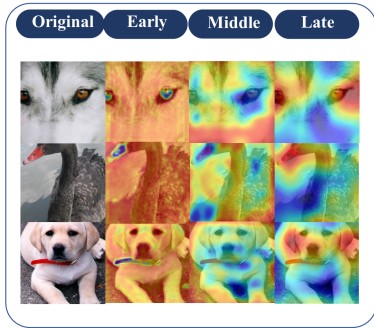 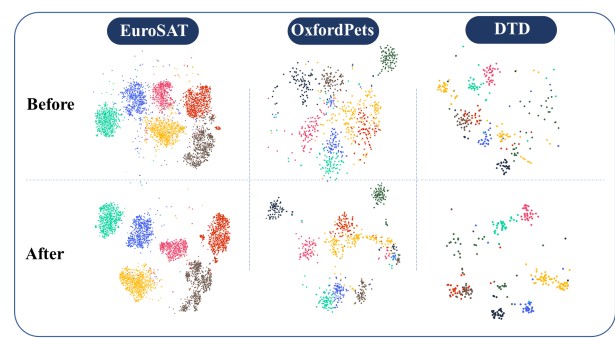

Figure 2: **Left:** Heatmap of the prompt's attention to the same image over different periods. Red represents more attention and blue represents less attention. **Right:** t-SNE plots of image embeddings of before and after image augmentation on EuroSAT, OxfordPets and DTD datasets. The distribution is denser after image augmentation than before.

label smoothing, thereby substantially increasing the model's complexity and potentially degrading performance (Seo et al., 2021). Consequently, achieving the optimal balance between accuracy and efficiency in large vision-language models for few-shot learning tasks presents a challenge.

To overcome these limitations, we introduce Multi-Vision Multi-Prompt (MVMP) method for few-shot learning tasks without additional parameters. MVMP uses *multi-prompt* to make predictions from multiple perspectives at different training stage. It also increases image-text diversity by *mixing self-augmentation* and *text distillation*. Using these three techniques can comprehensively strengthen the model's learning capabilities and achieve more accurate and efficient visual-text alignment.

Specifically, a) *Multi-prompt.* In Figure 2, during the early stages of training, prompts are more generalized, while in the later stages, prompts become more focused on specific details. To leverage the unique perspective of different stages of text prompts, we maintain multiple prompts at each training stage and perform information fusion through an average weighting method to achieve comprehensive improvement in model performance. b) *Mixed self-augmentation.* Considering the limited availability of image data in a few-shot learning environment, we propose the mixed self-augmentation framework. This framework entails replacing region pixels with pixels from another image region to create a new virtual image, enhancing image diversity and maintaining computational efficiency. In Figure 2, mixed self-augmentation optimizes class distributions in few-shot learning tasks. c) *Text distillation.* After enhancing the images, to address the problem of image-text mismatch caused by a single prompt, we utilize text distillation to acquire multiple text features from CLIP text encoder with fixed prompts (Khattak et al., 2023) which effectively expands and enhances the semantic representation of the text, further augmenting the alignment capabilities.

In this paper, we focus on adapting few-shot learning tasks. We compare MVMP with 10 state-of-the-art prompt learning and few-shot learning image augmentation baseline methods on 11 CLIP-based few-shot learning datasets. The experiment results demonstrate that our method improves overall accuracy by 2% to 4.6% while maintaining good generalization performance. In summary, our multi-vision multi-prompt method has the following main contributions:

- To further utilize the potential of each text prompt, we use a prompt memory bank to store prompts at different stages and leverage information fusion through average weight prediction.

- To increase the vision diversity in few-shot learning environment, we generate new virtual samples by using mixed self-augmentation, and stabilize the model through consistency loss.

- To align image-text pairs, we distill the prompt by obtaining multiple textual features through CLIP text encoder to include textual diversity and improve the robustness of the prompt.

## 2 RELATED WORKS

For a more detailed discussion of related works, please refer to Appendix A.1.

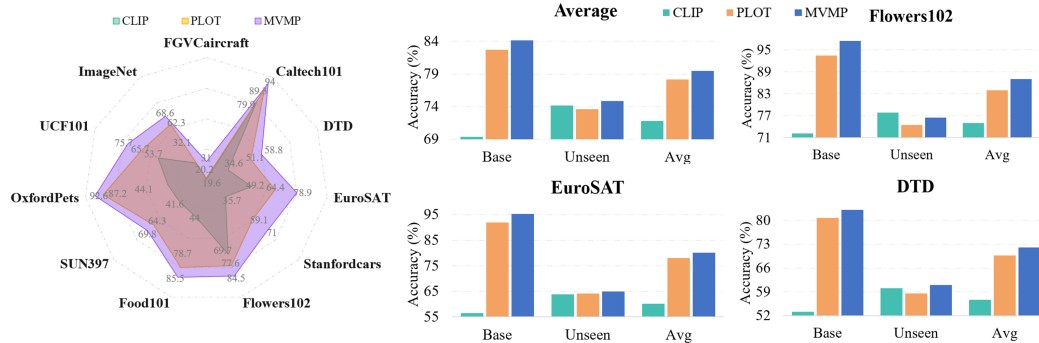

Figure 3: **Left:** Radar chart showcasing MVMP is superior performance in 1-shot image categorization across 11 diverse datasets compared with multiple prompt method. **Right:** Average results of base-to-new classification on 11 datasets. Base and New accuracies for MVMP are the highest on average and for most datasets, with more specific data detailed in the Appendix A.5

## 2.1 PROMPT LEARNING IN VISION-LANGUAGE MODELS

The typical pre-trained vision-language models learn in a self-supervised manner via text and image encoders and contrast loss (Radford et al., 2021; Jia et al., 2021). However, fine-tuning these models becomes challenging due to the large number of parameters. Recent studies have shown that the adaptation to certain tasks can be efficiently improved by adding prompts to the text encoder.

Prompt learning, first used in the field of NLP (Radford et al., 2019), has proven to be an efficient way of adapting large models to specific downstream tasks. Unlike fine-tuning, it uses text prompts rather than re-training model parameters (Devlin et al., 2019; Petroni et al., 2019). Recent studies such as Coop (Zhou et al., 2022b) and CoCoOP (Zhou et al., 2022a) use learnable prompts on CLIP with high accuracy to specific tasks. MaPLe (Khattak et al., 2022) and PromptSRC (Khattak et al., 2023) approaches make similar attempts by using high-level prompt and CLIP knowledge distillation. However, each of these methods uses a single prompt as a textual prompt, which has limitations for multi-category tasks. Learning multiple prompts in PLOT (Chen et al., 2023) by optimal transport, however, increases the optimization complexity and computational burden.

## 2.2 FEW-SHOT LEARNING IN VISION-LANGUAGE MODEL

In order to improve the learning capacity of few-shot learning, few-shot learning is usually addressed through meta-learning and data augmentation (Liu et al., 2021; Ravi & Larochelle, 2017). Meta-learning algorithms such as MAML (Finn et al., 2017), Reptile (Nichol et al., 2018) update parameters via gradient descent. But these methods increase computational cost in large vision-language model. Data augmentation like CutOut (Devries & Taylor, 2017), CutMix (Yun et al., 2019), SelfMix (Seo et al., 2021) and Mixup (Zhang et al., 2018) can result in problems such as information loss and complications with label smoothing (Ni et al., 2021). Therefore, we use a new mixed self-augmentation method to reduce the complexity and increase the image diversity.

## 3 METHODOLOGY

In contrast to the fine-tuning approach, prompt learning only trains prompt as the only learnable parameter (Zhou et al., 2022b; Khattak et al., 2023). The framework of MVMP is shown in Figure 4. Unlike previous approaches that use single prompt, our approach employs multi-stage prompts and image augmentation. Firstly, we obtain new virtual samples using mixed self-augmentation, replacing the random region with another image region. Secondly, consistency loss is used after image augmentation to stabilize the model's robustness. Then, the text feature of the prompt is influenced by the consistency loss of the textual distillation. Finally, text prompts are sequentially saved to the prompt memory during the training phase, and high-level prompts are assigned Gaussian weights based on performance. The final prediction integrates multi-stage prompt-weighted predictions.

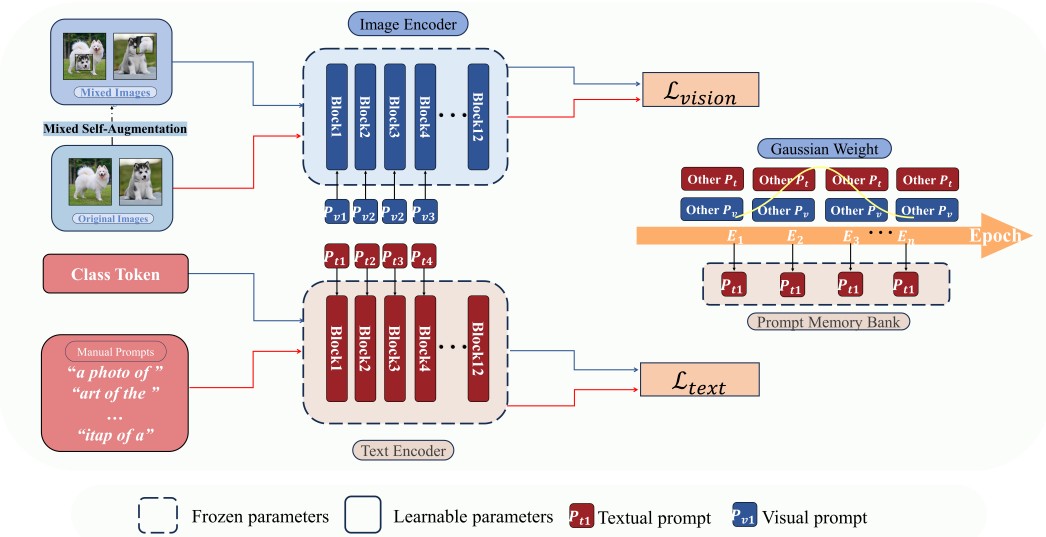

Figure 4: Overview of our multi-vision multi-prompt method (MVMP). MVMP generates new visual samples through mixed self-augmentation and performs the consistency loss with the original image. Then, it enhances the text samples using text distillation with consistency loss. Additionally, it stores the first layer prompt in the memory bank during each epoch and applies Gaussian weights to the high-layer prompts. Finally, integrate the prompts from different stages for average prediction.

## 3.1 Review of CLIP

CLIP consists of two encoders, and one is an image encoder which is based on ResNet (He et al., 2015) or ViT (Dosovitskiy et al., 2021). The other is a text encoder, which is based on Transformer (Vaswani et al., 2017). CLIP pre-trains the contrast loss between image and text to get a correspondence between image and text which improved performance on various vision tasks.

In the pre-training phase of the CLIP framework, CLIP utilizes a large set of image-text pairs to learn cross-modal feature representations, denoted as $\{(\boldsymbol{i}_1, \boldsymbol{t}_1), (\boldsymbol{i}_2, \boldsymbol{t}_2), \ldots, (\boldsymbol{i}_N, \boldsymbol{t}_N)\}$. For downstream image classification tasks, text prompts such as "a photo of a," are pre-concatenated to form the text input token sequence. Let the image encoder and text encoder be represented by $f(\theta_f)$ and $g(\theta_g)$, respectively. The text token sequence $\boldsymbol{T}$ is formulated as a concatenation of various elements: the prefix, the prompt, the [CLS] token, and the suffix. Specifically, if the length of the prompt is $l$, then the text token sequence $\boldsymbol{T}$ can be represented as: $\boldsymbol{T} = \{\boldsymbol{t}_{\text{prefix}}, \boldsymbol{t}_1, \boldsymbol{t}_2, \ldots, \boldsymbol{t}_l, \boldsymbol{CLS}, \boldsymbol{t}_{\text{suffix}}\}$. Here, $\boldsymbol{t}_l$ is prompt sequence while $\boldsymbol{t}_{\text{prefix}}$ and $\boldsymbol{t}_{\text{suffix}}$ are the prefix and suffix tokens respectively. The [CLS] token serves as each class label. The text feature from the text encoder is $g(\boldsymbol{T}, \theta_g)$. Similarly, the image input is divided into $n$ patches in transformer embedding, which is $\boldsymbol{I} = \{\boldsymbol{i}_1, \boldsymbol{i}_2, \ldots, \boldsymbol{i}_n\}$. And image feature is represented as $f(\boldsymbol{I}, \theta_f)$. The inference process for input image x with class k is :

$$p(y = k|\mathbf{x}) = \frac{\exp\left(\text{sim}\left(f(\boldsymbol{I}, \theta_f), g(\boldsymbol{T}, \theta_g)\right)/\tau\right)}{\sum_{k'=1}^{N} \exp\left(\text{sim}\left(f(\boldsymbol{I}, \theta_f), g(\boldsymbol{T}_{k'}, \theta_g)\right)/\tau\right)} \tag{1}$$

where, $\text{sim}(\cdot, \cdot)$ represents the similarity function between the image and text feature vectors.

**Prompt Learning for CLIP.** At this stage, CLIP prompt research refers to embedding prompts in the image or text encoder. The structure of the baseline independent vision-language prompt (Rasheed et al., 2023) used in this paper is a multilayer prompt embedding. To learn deeper prompts, we use a deep structure to learn prompts at each transformer block. Specifically, The text encoder utilizes learnable prompt set $\boldsymbol{P}_t$, which contians each layer text prompt $\boldsymbol{P}_t = [\boldsymbol{p}_{t1}, \boldsymbol{p}_{t2}, \ldots, \boldsymbol{p}_{tn}]$ for $n$ layers. The text encoder input is thus represented as $\boldsymbol{T}_p = \{\boldsymbol{t}_{\text{prefix}}, \boldsymbol{P}_t, \boldsymbol{CLS}, \boldsymbol{t}_{\text{suffix}}\}$. Similarly, the image encoder features a learnable prompt $\boldsymbol{P}_v$, represented as $\boldsymbol{P}_v = [\boldsymbol{p}_{v1}, \boldsymbol{p}_{v2}, \ldots, \boldsymbol{p}_{vn}]$ for $n$ layers. The input image is represented as $\boldsymbol{I}_p = \{\boldsymbol{P}_v, \boldsymbol{i}_1, \boldsymbol{i}_2, \ldots, \boldsymbol{i}_n\}$. Consequently, the image feature and text feature are computed as $f(\boldsymbol{I}_p, \theta_f)$ and $g(\boldsymbol{T}_p, \theta_g)$. When the model is applied to a downstream target

dataset $\mathcal{D}$, the text and image encoders of CLIP are frozen, and only the embedded text and image prompts $\boldsymbol{P}_t$ and $\boldsymbol{P}_v$ are trained. The optimization of these prompts is carried out by minimizing the cross-entropy loss function for image data $\boldsymbol{x}$ and class token $\boldsymbol{y}$:

$$
\begin{aligned}
\mathcal{L}_{\text{ce}} &= -\sum_{(\boldsymbol{I}_p, \boldsymbol{T}_p) \in \mathcal{D}} \log \left( \frac{\exp(\text{sim}(f(\boldsymbol{I}_p, \theta_f), g(\boldsymbol{T}_p, \theta_g))/\tau)}{\sum_{T'_p \in \mathcal{D}} \exp(\text{sim}(f(\boldsymbol{I}_p, \theta_f), g(T'_p, \theta_g))/\tau)} \right) \\
&= -\sum_{(\mathbf{x}, \mathbf{y}) \in \mathcal{D}} \log \left( \frac{\exp(\text{sim}(f(\mathbf{x}, \theta_f, \boldsymbol{P}_v), g(\mathbf{y}, \theta_g, \boldsymbol{P}_t))/\tau)}{\sum_{y' \in \boldsymbol{T}_p} \exp(\text{sim}(f(\mathbf{x}, \theta_f, \boldsymbol{P}_v), g(y', \theta_g, \boldsymbol{P}_t))/\tau)} \right)
\end{aligned}
\tag{2}
$$

## 3.2 Multiple Prompts

Although a multilayered prompt structure is employed to gradually learn different features from the abstract to the concrete level, each stage of training utilizes prompts with distinct capabilities. As illustrated in Figure 2, prompts in the early stages aim to capture more generalized features, whereas prompts in later stages increasingly focus on capturing detailed features. Relying solely on late-stage detail prompts limits the model's generalization ability in more diverse situations. An intuitive solution to this issue is combining all prompts from training stages with different perspectives.

**Prompt Memory Bank.** To use the specific capabilities of each stage, we separately store the prompts associated with each stage. After completing each training epoch, we update our prompt memory bank $M$. This memory bank $M$ contains all prompts generated from the start of the training until the current epoch. To update $M$ for a prompt $\boldsymbol{p_e}$ generated in epoch $e$:

$$
M_{e+1} = M_e \cup \{\boldsymbol{p_e} \cdot I(e)\}, \text{ where } I(e) = \begin{cases} 1 & \text{if } \epsilon_e > \beta \\ 0 & \text{otherwise} \end{cases}
\tag{3}
$$

where, $\epsilon_e$ represents the evaluation parameter, and $\beta$ is a threshold that determines which prompts should be included in $M$. $e$ denotes the total number of training epochs. This updating strategy ensures that the model's understanding is continuously captured throughout the training process.

**Gaussian Weights for High-layer Prompts.** High-layer prompts tend to be more abstract and detail-oriented (Khattak et al., 2023). While there are advantages to retaining all layers of prompts, this also increases the memory load. Therefore, we choose to store only the text prompt from the first layer in the memory bank. During training, Gaussian weights were assigned to all other high-layer prompts, excluding the first layer, to enable efficient weighted combinations. This approach allows us to fully exploit each prompt layer without adding extra memory burden. Specifically, the representation for the weighted aggregation of the high-layer prompts in $H$ during the $e$-th epoch is:

$$
\boldsymbol{P_w}^{(e)} = \sum_{i \in H} w_i^{(e)} \boldsymbol{P_i}
\tag{4}
$$

where, $\boldsymbol{P_w}^{(e)}$ is the weighted prompts at the $e$-th epoch, $\boldsymbol{P_i}$ represents the prompt at the $i$-th layer, $w_i^{(e)}$ is the corresponding Gaussian weight at the $e$-th epoch, and $H$ contains all high-layer prompts.

With the assistance of the prompt memory bank and Gaussian weights, we are able to overcome some of the fundamental limitations of prompt learning and obtain a wealth of meaningful prompt information. The prompt memory bank can accommodate and effectively utilize a diverse array of useful prompts, enhancing the model's flexibility and adaptability during the learning process. The Gaussian weighting mechanism for high-layer prompts enables more precise utilization of their predictive power from multiple layers, resulting in comprehensive and accurate predictions.

## 3.3 Vision and Text Augmentation

**Mixed Self-Augmentation.** To address the issue of limited image information available for the few-shot learning task, we propose a novel framework called mixed self-augmentation. This framework is specifically designed to quickly acquire diverse and rich image features. To accomplish this, we generate new mixed virtual samples $\boldsymbol{I}_{\text{mixed}}$ by randomly selecting a region from one input sample

$I_1$ and replacing its pixels with the corresponding region from another sample $I_2$ (Seo et al., 2021). Specifically, given input images $I_1$ and $I_2$ with dimensions $W \times H$, a source sub-area $P1$ is randomly selected from $I_1$ with dimensions $rw \times rh$, and its upper-left corner coordinates $(ra_1, rb1)$ are drawn from a uniform distribution as follows:

$$(ra_1, rb_1) \sim U\left([0, W - rw] \times [0, H - rh]\right), \text{ where } U(x; a, b) = \begin{cases} \frac{1}{b-a} & \text{for } a \leq x < b, \\ 0 & \text{otherwise.} \end{cases} \quad (5)$$

Similarly, a target subregion $P2$ is randomly selected from $I_2$ with upper-left corner coordinates $(ra_2, rb_2)$ satisfying:

$$(ra_2, rb_2) \sim U\left([0, W - rw] \times [0, H - rh]\right) \quad (6)$$

A pixel substitution operation is performed by replacing $P1$ with $P2$, thus generating a new virtual sample $I_{\text{mixed}}$ satisfying:

$$I_{\text{mixed}}[ra_1 : ra_1 + rw, rb1 : rb1 + rh] = I_2[ra_2 : ra_2 + rw, rb2 : rb2 + rh] \quad (7)$$

This method replaces region images with different values by culling them, effectively scrambling the data. This strategy helps prevent the network from over-fitting to the existing samples and ultimately improves its generalization performance.

**Consistency Loss.** To guarantee the model's stability, we introduce a consistency loss applied to both the original and mixed images. Specifically, the original image is represented as $\mathbf{x}$, and the mixed image is denoted by $\mathbf{x}'$. The cross-entropy losses calculated for these images are $\mathcal{H}(\mathbf{x})$ and $\mathcal{H}(\mathbf{x}')$, respectively. The consistency loss is defined as:

$$\mathcal{L}_{\text{vision}} = \left(\mathcal{H}(\mathbf{x}) - \mathcal{H}(\mathbf{x}')\right)^2 \quad (8)$$

**Text Distillation.** After increasing image diversity, an important challenge that restricts the learning capability of the prompt is the presence of only one text feature per category, resulting in a diversity mismatch between the image and text modalities. To address this disparity, we propose a method of text feature distillation by leveraging multiple fixed prompts.

This method aims to enhance textual diversity by using CLIP text encoders to distill textual features from a variety of fixed prompts and use them as a guide to learnable prompts. We define the loss function $\mathcal{L}_{\text{text}}$ for this distillation process as follows:

$$\mathcal{L}_{\text{text}} = \frac{1}{N} \sum_{i=1}^{N} \left(g(\mathbf{y}_i, \theta_g, P_{\text{fixed}}) - g(\mathbf{y}_i, \theta_g, P_t)\right)^2 \quad (9)$$

where $P_{\text{fixed}}$ and $P_t$ are the fixed prompts and the learnable prompt, respectively. In this method, we utilize fixed prompt to enhance text features through distillation, absorbing characteristics from various contexts, thus overcoming the constraint of relying solely on a single text feature. This approach not only enhances the model's learning efficiency but also augments its overall performance.

**Overall Optimization.** In our model, the overall optimization method is:

$$\mathcal{L} = \lambda_1 \mathcal{L}_{\text{vision}} + \lambda_2 \mathcal{L}_{\text{text}} + \lambda_3 \mathcal{L}_{\text{ce}} \quad (10)$$

where, $\lambda$ is weight coefficient. By enhancing the diversity of images and texts, the integration of the three loss functions has enhanced the model's performance on individual tasks and improved its robustness and adaptability across various data types. For a more comprehensive overview of the algorithm, please refer to the training and testing processes pseudo-code provided in Appendix A.4.

## 4 EXPERIMENTS

To evaluate our approach, we conduct extensive experiments including a comparison with CLIP's prompt learning state-of-the-art approach and few-shot learning state-of-the-art approach in the few-shot learning task, cross-dataset studies and ablation studies.

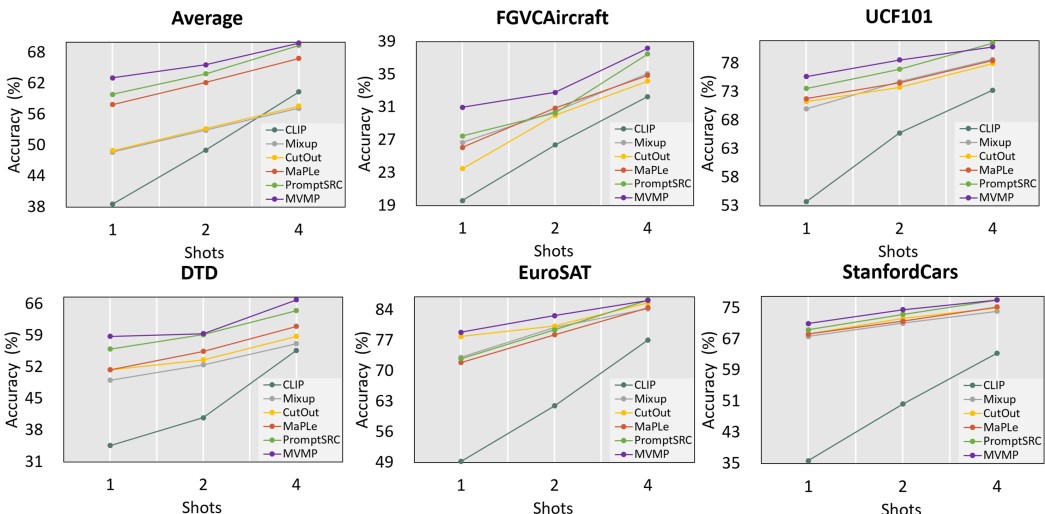

Figure 5: Comparison of accuracy in few-shot environment for shot 1, 2, 4 across 5 datasets. MVMP has a great improvement in training with fewer samples, *i.e.* shot=1.

## 4.1 EXPERIMENT SETTINGS

**Benchmark Settings.** We perform two types of benchmark experiments. First, we perform few-shot learning experiments comparing with CLIP's prompt learning methods and traditional few-shot learning data augmentation methods. Second, we also perform cross-dataset benchmark. Addtionally, we perform ablation experiments, comparing the number of prompts and MVMP settings.

**Datasets.** First, following the experimental design of the CLIP model in the evaluation of few-shot learning, we evaluate the performance of 11 different visual categorization datasets including ImageNet (Deng et al., 2009), OxfordPets (Parkhi et al., 2012), StanfordCars (Krause et al., 2013), Caltech101 (Fei-Fei et al., 2007), DTD (Cimpoi et al., 2014), EuroSAT (Helber et al., 2019), FGV-CAircraft (Maji et al., 2013), Flowers102 (Nilsback & Zisserman, 2008), Food101 (Bossard et al., 2014), SUN397 (Xiao et al., 2010) and UCF101 (Soomro et al., 2012), which cover a wide range of datasets from large-scale image classification (e.g., ImageNet) to more specialized and fine-grained tasks (e.g., pet, car, flower, and food categorization), as well as texture, land-cover, scene, and action recognition. For a more detailed description of the dataset, please refer to Appendix A.6.

**Baselines.** We totally compare with 10 state-of-the-art baselines. For models based on CLIP prompt learning, we perform comparisons with state-of-the-art methods, which include CLIP (Radford et al., 2021), CoOp (Zhou et al., 2022b) and CoCoOP (Zhou et al., 2022a) for the first introduction of prompt learning, PLOT (Chen et al., 2023) for multi-prompt learning, as well as Maple (Khattak et al., 2022) and PromptSRC (Khattak et al., 2023)for multilayer structures. In terms of traditional few-shot learning data augmentation, we compare with CutOut (Devries & Taylor, 2017), CutMix (Yun et al., 2019), SelfMix (Seo et al., 2021) and Mixup (Zhang et al., 2018).

**Implementation Details.** For experiments on few-shot learning, we use a similar setup to that few-shot learning in CLIP. Specifically, use 1, 2, 4, 8 and 16 shots as training set and evaluate with the original test set. For cross-dataset benchmark, we perform 16 shots training on ImageNet and evaluate it on a full test set of the other 10 different datasets. For ablation experiments, we use 1 shot for training and test on 11 datasets to get average results. In our experiments, VIT-B/16 is used as the image encoder for CLIP and all results are based on the average of 3 times. We use 5 prompts from memory bank for prediction. For more detailed information please refer to Appendix A.6.

## 4.2 FEW-SHOT EXPERIMENT RESULTS

We first experimentally evaluate CLIP prompt learning and data augmentation methods on 11 specific few-shot learning datasets dedicated to CLIP evaluation. The results of these evaluations are reported in Table 1 and Table 2. It is evident from the tables that our proposed method, MVMP,

Table 1: Accuracy of classification for **CLIP prompt learning** methods on 11 datasets in **1-shot** adaptation task. The **bold** number indicates the best result.

| Method | Aircraft | Caltech | DTD | EuroSAT | Cars | Flowers | Food | SUN397 | Pets | UCF | ImgNet | Average |
|---|---|---|---|---|---|---|---|---|---|---|---|---|
| **CLIP** | 19.6 | 79.9 | 34.6 | 49.2 | 35.7 | 69.7 | 44.0 | 41.6 | 44.1 | 53.7 | 32.1 | 45.8 |
| **CoOp** | 21.4 | 92.6 | 49.0 | 53.4 | 67.4 | 77.5 | 84.3 | 66.8 | 90.4 | 71.2 | 66.3 | 67.3 |
| **CoCoOp** | 12.7 | 93.8 | 48.5 | 55.3 | 67.2 | 72.1 | 85.3 | 68.3 | 91.3 | 70.3 | 67.4 | 66.6 |
| **PLOT** | 27.7 | 93.1 | 54.0 | 64.4 | 68.1 | 80.0 | 78.7 | 66.0 | 91.1 | 73.4 | 66.4 | 69.4 |
| **MaPLe** | 26.1 | 92.9 | 51.4 | 74.1 | 68.3 | 83.3 | 79.9 | 64.8 | 89.1 | 71.8 | 62.7 | 69.5 |
| **PromptSRC** | 27.5 | 93.2 | 56.7 | 73.1 | 69.4 | **85.1** | 84.9 | 69.0 | 91.5 | 73.6 | 67.7 | 71.9 |
| **MVMP(Ours)** | **31.0** | **94.0** | **58.8** | **78.9** | **71.0** | 84.5 | **85.5** | **69.8** | **92.6** | **75.7** | **69.1** | **73.8** |
| Δ | +3.5 | +1.0 | 2.8 | +6.1 | +1.6 | -0.4 | 0.7 | +0.4 | +0.9 | +2.1 | +1.1 | +1.9 |

Table 2: Accuracy of classification for **few-shot learning data augmentation** methods on 11 datasets in **1-shot** adaptation task. The **bold** number indicates the best result.

| Method | Aircraft | Caltech | DTD | EuroSAT | Cars | Flowers | Food | SUN397 | Pets | UCF | ImgNet | Average |
|---|---|---|---|---|---|---|---|---|---|---|---|---|
| **CLIP** | 19.6 | 79.9 | 34.6 | 49.2 | 35.7 | 69.7 | 44.0 | 41.6 | 44.1 | 53.7 | 32.1 | 45.8 |
| **+CutMix** | 24.7 | 92.7 | 47.8 | 51.7 | 59.0 | 84.3 | 81.4 | 62.3 | 86.2 | 70.9 | 61.5 | 65.7 |
| **+SelfMix** | 27.1 | 93.4 | 51.9 | 66.7 | 52.8 | 73.7 | 83.7 | 66.4 | 90.1 | 69.0 | 65.5 | 67.3 |
| **+Mixup** | 26.7 | 93.5 | 49.1 | 73.1 | 67.7 | 75.1 | 82.0 | 66.0 | 90.0 | 70.0 | 66.1 | 69.0 |
| **+CutOut** | 23.5 | 92.6 | 51.4 | 77.9 | 68.3 | 75.6 | 82.5 | 65.0 | 89.1 | 71.3 | 62.7 | 69.1 |
| **MVMP(Ours)** | **31.0** | **94.0** | **58.8** | **78.9** | **71.0** | **84.5** | **85.5** | **69.8** | **92.6** | **75.7** | **69.1** | **73.8** |
| Δ | +7.5 | +1.4 | +7.4 | +1.0 | +2.7 | +8.9 | +3.0 | +4.8 | +3.5 | +4.4 | +5.9 | +4.6 |

outperforms other approaches on almost all of the datasets in the challenging task of extreme 1-shot learning. Compared to advanced data augmentation methods for few-shot learning, we observe a 4.6% improvement. Against PromptSRC, which represents the current state-of-the-art in prompt learning, MVMP achieves a 2% increase in accuracy. Specifically, MVMP demonstrates superior performance on 10/11 tested datasets. This approach excels higher when dealing with datasets that exhibit a high level of diversity and complexity such as ImageNet and DTD. MVMP consistently delivers excellent results across various datasets and complexities through a straightforward approach involving multiple images and prompts. This demonstrates its wide applicability and robustness.

In Figure 5, we choose five datasets with more than 10% variation in accuracy for comparison. When the number of training samples is minimal, the performance of MVMP demonstrates a noticeable improvement. This improvement can be attributed to greater diversity, enabling the model to capture various visual features and semantic information more effectively. Therefore, leveraging diversity as a strategy has proven effective in scenarios where the number of samples is scarce. For more detailed experiment results on few-shot learning (including 2, 4, 8, 16 shots), please refer to Appendix A.3.

## 4.3 CROSS-DATASET EXPERIMENT RESULTS

In Table 3, we show the performance comparison of cross-dataset. Using the same settings as the previous method, we utilize 16 shots of ImageNet as the source dataset for training and test on 10 other datasets. The results show that MVMP performs with highest accuracy on the source dataset, outperforming all prompt learning methods. Meanwhile, it maintains good generalization performance on 7/10 datasets, while the performance on the remaining 3 is comparable to the top methods. This demonstrates that MVMP exhibits reliable generalization performance. Additionally, we compare the results of base-to-new experiment for 11 datasets, as detailed in the Appendix A.5.

## 4.4 ABLATION EXPERIMENT RESULTS AND COMPLEXITY COMPARISON

In this section, we conduct ablation experiments for different components. As shown in Figure 6, Firstly, compared to using a single prompt, the use of multiple prompts led to a 4% improve-

Table 3: Cross-dataset benchmark evaluation. The **bold** number indicates the best result.

| | Source | Target | | | | | | | | | | |
|---|---|---|---|---|---|---|---|---|---|---|---|---|
| | Imagenet | Aircraft | Caltech | DTD | EuroSAT | Cars | Flowers | Food | SUN397 | Pets | UCF101 | Average |
| **CoOp** | 71.5 | 18.5 | 93.7 | 42.0 | 46.4 | 64.5 | 68.7 | 85.3 | 64.2 | 89.1 | 66.6 | 63.9 |
| **CoCoOp** | 71.0 | 22.9 | **94.4** | 45.7 | 45.4 | 65.3 | 71.1 | 86.1 | 67.4 | 90.6 | 68.2 | 65.7 |
| **MaPLe** | 70.7 | **24.7** | 93.3 | 46.2 | **48.0** | 65.3 | 71.6 | 86.2 | 67.0 | 90.5 | 68.4 | 66.1 |
| **PromptSRC** | 71.3 | 23.9 | 93.6 | 46.9 | 45.5 | 65.7 | 70.2 | 86.2 | 67.1 | 90.3 | 68.8 | 65.8 |
| **MVMP** | **72.7** | 23.5 | 93.5 | **47.2** | 47.7 | **65.8** | **71.3** | **86.3** | **67.6** | 90.6 | **69.2** | **66.3** |

| Setting | Multi-Prompt | Multi-vision | Text diversity | Avg |
|---|---|---|---|---|
| | ✗ | ✗ | ✗ | 67.3 |
| | ✓ | ✗ | ✗ | 71.3 |
| MVMP | ✓ | ✓ | ✗ | 71.7 |
| | ✓ | ✓ | ✓ | **73.8** |

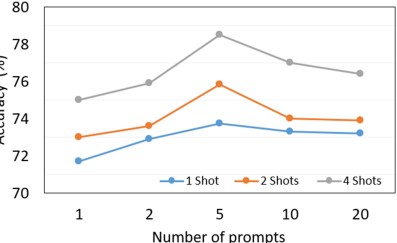

Figure 6: **Left:** MVMP ablation experiments on 1-shot 11 datasets for different settings. **Right:** Accuracy in 11 datasets for 1, 2, 4 shots task with different number of prompts

ment in accuracy. Furthermore, incorporating multiple visions contributes to an additional 0.5% enhancement. This improvement is mainly due to the enhancement of only the image part, resulting in a mismatch between image and text. After increasing text diversity, the accuracy increases by 2%. Secondly, we also experiment with the number of prompts stored in the prompt memory bank. Among the options of 1, 2, 5, 10, and 20 prompts, we find that the accuracy is highest when using 5 prompts, while using a single prompt is the least effective. This suggests that it is challenging for a single prompt to account for both the generalization and the task details.

In contrast, using multiple prompts from different periods allows for a more comprehensive capture of the complexity of the task. Details of the prompt length ablation experiment is in the Appendix A.2. Additionally, we compare the training times of CoCoOp, PLOT, and MVMP by training and testing them on the DTD dataset in a 1-shot setting. As shown in Table 4, MVMP is 32.5% faster in training compared to the PLOT. Additionally, it boasts a performance advantage, with up to a 17.4% increase in accuracy over the meta-learning approach, CoCoOp.

Table 4: Comparison of training time and accuracy on the DTD dataset.

| Method | Training (s) | Acc. (%) |
|---|---|---|
| CoCoOp | 137 | 49.5 |
| PLOT | 280 | 54.7 |
| MVMP | 189 | 58.1 |

## 5 CONCLUSION AND FURTHER STUDY

This study presents a strategy called Multi-Vision Multi-Prompt (MVMP) to optimize prompt storage at different stages, aiming for a more comprehensive and diverse selection. By increasing image and text diversity, MVMP enhances prompt learning and improves the model's robustness without introducing additional parameters. Our method demonstrates significant enhancements in CLIP's prompt learning and advanced image augmentation methods. We observe notable performance improvements through various few-shot learning tasks across multiple datasets and it maintains good generalization. This validates the effectiveness of the multi-vision, multi-prompt strategy and highlights the superiority of using multiple prompts over a single one. While MVMP has shown significant advantages and usefulness across various domains, it is essential to investigate its efficiency and generalization ability thoroughly. In our future research, we will focus on improving the efficiency of MVMP's inference process to strike a better balance between accuracy and efficiency. Additionally, there is still room for further optimizing MVMP regarding its generalization ability.

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

# A APPENDIX

The following is supplemental material that includes a more detailed description of the related work, additional implementation details, a description of the dataset, a description of the algorithm, full experimental results for few-shot Learning, a comparison for prompt length, results for base-to-new experiments and compression of complexity. The table of contents is as follows:

## A.1 RELATED WORKS

### A.1.1 VISION LANGUAGE PRE-TRAINED MODEL

The Vision-Language Pre-trained (VLP) models aim to establish cross-modal connections through extensive pre-training, incorporating both visual and textual modalities. Typical Vision-Language Pre-trained models consist of text and image encoders, utilizing self-supervised learning through contrast loss (Radford et al., 2021; Jia et al., 2021; Gao et al., 2021). While Vision-Language Pre-trained models such as CLIP have strong linguistic parsing capabilities and can be adapted to a wide range of specific downstream tasks (Ding et al., 2022a; Gu et al., 2022; Li et al., 2022a; Rasheed et al., 2022), it is problematic to maintain their generalizability when applied to these tasks. Recent research has explored more effective ways to adapt VLP to downstream tasks, namely by incorporating prompts into text encoder (Wang et al., 2023). This approach preserves the original capabilities of models and demonstrates their potential advantages for specific tasks.

### A.1.2 PROMPT LEARNING

Prompt learning was initially developed by the natural language processing (NLP) field as a means to effectively tailor large-scale models for downstream tasks (Radford et al., 2019; Liu et al., 2023; Ding et al., 2022b). Unlike fine-tuning methods, prompt learning does not involve re-training the model parameters but instead uses text prompts to guide large models to specific tasks. In the case of a pre-trained language model, the prompt is typically presented in the form of a completion or masked sentence, such as "*It was [MASK]*" The model is trained to predict the appropriate word for the masked position (Devlin et al., 2019; Petroni et al., 2019; Luo et al., 2021). Most early studies manually set prompts, a time-consuming and challenging task for finding the appropriate prompt. In contrast, recent research focuses on enabling the model to learn more suitable prompts independently (Zhang et al., 2020; 2022). The CoOp (Zhou et al., 2022b)and CoCoOP (Zhou et al., 2022a) two approaches first use a learnable prompt based on the CLIP modeling. The former uses learnable prompts to enhance the CLIP model's adaptation to specific tasks while preserving its generalization ability. The latter adds image biases to the prompts for further improving the CLIP model's generalization performance. On the other hand, the PLOT method (Chen et al., 2023), attempts to divide the image into multiple region prompts and learns these prompts using the optimal transport strategy. Maple method (Khattak et al., 2022) adds prompts in deeper layers of the model to ensure that these prompts are also useful in high-level abstract feature representations. These approaches explore how the prompt enhances CLIP from various dimensions. However, they frequently have the limitation of using a single prompt to represent the entire dataset, even though different categories may require different prompts. In this regard, the idiosyncrasies between different categories are disregarded.

### A.1.3 FEW-SHOW LEARNING IN VISION-LANGUAGE MODELS

Large Vision-Language Models can adapt quickly in few-shot learning application scenarios due to large-scale pre-training. Few-shot learning is usually achieved by training with only a tiny number (typically no more than 5) of samples for each classification, thus achieving the goal of accurately classifying a more significant number of samples (Li et al., 2022b; Cubuk et al., 2018; Vinyals et al., 2016; Chen et al., 2021; Fang et al., 2023; Sreenivas & Biswas, 2023). Traditionally, this type of problem has been solved in two main ways: meta-learning and data augmentation.

For meta-learning (Lee et al., 2023), there are algorithms such as MAML (Finn et al., 2017), Reptile (Nichol et al., 2018), etc., which update all the parameters in the network by gradient descent during multiple model fine-tuning stages. However, in large vision language models, applying meta-learning (e.g., CoCoOp (Zhou et al., 2022a) usually requires the introduction of additional parameters. This not only increases the computational burden but also makes parameter updating more complicated. Therefore, to reduce the computational cost and the complexity of parameter management, we prefer to use data augmentation methods to improve the model's performance in few-shot environments. Data augmentation method includes CutOut (Devries & Taylor, 2017) randomly cuts the area image as a new image, CutMix (Yun et al., 2019) mixes the cut area into another image, SelfMix (Seo et al., 2021) does region replacement on the same image and Mixup (Zhang et al., 2018) mixes different images on top of each other to form a new image. However, these methods can result in problems such as information loss and complications with label smoothing (Ni et al., 2021).

### A.2 ABLATION EXPERIMENT

In this section, we conduct ablation experiment on prompt length for 1 shot on 11 datasets, exploring prompt lengths of 1, 2, 4, 8, and 16. Figure 7 demonstrates the average results achieved on Imagenet as well as 11 other datasets. The findings indicate that the highest accuracy rate is attained when the length of the prompt word consists of 4 words. If there are either too few or too many prompt words, the accuracy rate diminishes. This phenomenon can be attributed to the fact that an excessive number of prompt words distracts the model, while an insufficient number of prompt words fails to provide adequate information to the model.

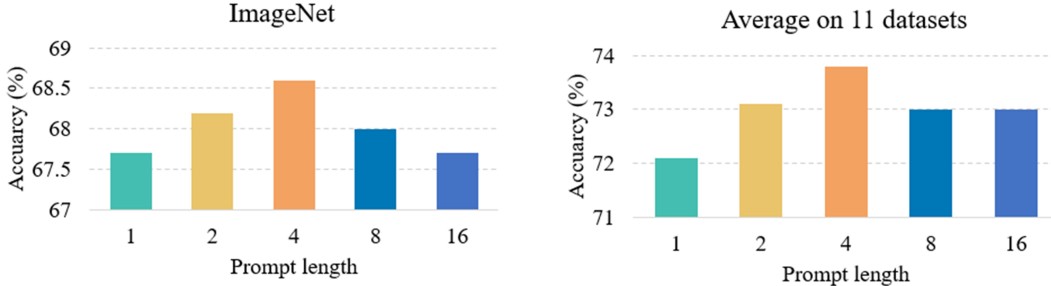

Figure 7: Ablation experiment for prompt length of 1, 2, 4, 8, 16 on ImageNet dataset and 11 datasets. When prompt length is 4, the accuracy achieves the highest.

### A.3 FEW-SHOT LEARNING EXPERIMENT

In this section, we present detailed results of few-shot learning experiments for shots 2, 4, 8, and 16 across 11 datasets. Table 5 shows the performance of 10 baselines in few-show learning of shots 2, 4, 8, 16. Out of all the state-of-the-art methods, MVMP consistently outperforms them in every experiment. Specifically, MVMP demonstrates remarkable performance improvements when applied to complex and challenging datasets such as ImageNet, DTD, and EuroSAT. Notably, the performance improvement achieved by MVMP is even more significant when the number of training samples is limited to shot=2 compared to shot=4. Overall, MVMP achieves the highest average accuracy across all shot experiments and exhibits improvements on nearly all datasets.

Table 5: Few-shot learning experiment of shot 2, 4, 8, 16 for 11 datasets. The **bold** number indicates the best result.MVMP achieves the highest accuracy in almost every dataset in every SHOT experiment, and the highest average accuracy as well.

| | Method | Craft | Calte. | DTD | Euro. | Cars | Flo. | Food | SUN | Pets | UCF | ImgNet | Average |
|---|---|---|---|---|---|---|---|---|---|---|---|---|---|
| **2 SHOTS** | CLIP | 26.4 | 89.0 | 40.8 | 62.0 | 50.3 | 85.1 | 61.5 | 53.7 | 58.4 | 65.8 | 44.9 | 58.0 |
| | CoOp | 26.2 | 93.1 | 53.6 | 65.2 | 70.5 | 87.3 | 84.4 | 66.5 | 89.8 | 73.4 | 67.1 | 70.6 |
| | CoCoOp | 15.1 | 94.8 | 52.2 | 46.7 | 68.4 | 75.8 | 86.2 | 69.0 | 92.6 | 73.5 | 69.8 | 67.6 |
| | PLOT | 30.1 | 94.0 | 56.0 | 76.5 | 72.2 | 30.1 | 82.5 | 68.0 | 91.3 | 72.6 | 68.3 | 67.4 |
| | MaPLe | 30.9 | 94.0 | 55.5 | 78.3 | 71.6 | 88.9 | 81.5 | 67.1 | 90.9 | 74.6 | 65.1 | 72.6 |
| | PromptSRC | 30.3 | 94.5 | 59.2 | 79.4 | 73.4 | 90.7 | 85.7 | 71.6 | 92.5 | 77.0 | 69.1 | 74.9 |
| | CutMix | 25.0 | 92.8 | 53.2 | 63.8 | 68.9 | 89.4 | 79.6 | 66.2 | 84.2 | 73.1 | 68.3 | 69.5 |
| | SelfMix | 28.3 | 93.8 | 51.1 | 65.6 | 67.2 | 82.5 | 82.8 | 68.8 | 89.3 | 73.2 | 68.3 | 70.1 |
| | Mixup | 30.5 | 94.2 | 52.5 | 80.0 | 71.1 | 85.8 | 80.9 | 67.8 | 90.2 | 74.8 | 67.4 | 72.3 |
| | CutOut | 30.0 | 94.6 | 53.6 | 80.3 | 70.6 | 84.9 | 82.5 | 68.5 | 90.2 | 73.8 | 68.9 | 72.5 |
| | **MVMP(Ours)** | **32.8** | **94.8** | **59.4** | **82.7** | **74.6** | **90.8** | **85.8** | **72.0** | **92.7** | **78.6** | **70.0** | **75.8** |
| **4 SHOTS** | CLIP | 32.3 | 92.1 | 55.7 | 77.1 | 63.4 | 92.0 | 73.2 | 63.0 | 71.2 | 73.3 | 54.9 | 68.0 |
| | CoOp | 30.8 | 94.4 | 58.7 | 70.8 | 74.5 | 92.2 | 84.5 | 70.0 | 92.6 | 77.1 | 68.7 | 74.0 |
| | CoCoOp | 24.8 | 95.0 | 55.0 | 65.6 | 69.4 | 78.4 | 86.9 | 70.2 | 92.8 | 74.8 | 70.4 | 71.2 |
| | PLOT | 25.3 | 95.0 | 62.4 | 83.2 | 75.2 | 92.3 | 83.0 | 71.7 | 92.0 | 79.2 | 63.9 | 74.8 |
| | MaPLe | 34.9 | 94.4 | 61.0 | 84.5 | 75.3 | 92.7 | 81.8 | 70.7 | 91.9 | 78.5 | 67.7 | 75.8 |
| | PromptSRC | 37.5 | 95.3 | 64.5 | **86.3** | 77.1 | 93.4 | 86.2 | 74.0 | 93.4 | **81.6** | 71.1 | 78.2 |
| | CutMix | 34.1 | 94.8 | 61.6 | 82.6 | 74.3 | 92.0 | 81.2 | 70.5 | 88.8 | 76.7 | 69.9 | 75.1 |
| | SelfMix | 34.7 | 95.2 | 57.1 | 75.9 | 69.8 | 89.5 | 83.8 | 71.6 | 91.8 | 76.4 | 69.9 | 74.2 |
| | Mixup | 35.1 | 95.0 | 57.2 | 84.3 | 74.2 | 82.4 | 82.8 | 71.2 | 92.2 | 78.7 | 68.6 | 74.7 |
| | CutOut | 34.2 | 94.8 | 58.8 | 85.6 | 73.6 | 90.0 | 82.8 | 71.1 | 92.2 | 78.0 | 70.0 | 75.6 |
| | **MVMP(Ours)** | **38.2** | **95.6** | **66.9** | 86.2 | **77.1** | **93.5** | **86.2** | **74.0** | **93.6** | 80.9 | **71.3** | **78.5** |
| **8 SHOTS** | CLIP | 39.4 | 93.4 | 63.5 | 84.4 | 73.7 | 96.1 | 44.0 | 69.1 | 78.4 | 79.3 | 62.2 | 71.2 |
| | CoOp | 39.0 | 94.4 | 64.8 | 78.1 | 79.3 | 95.0 | 84.3 | 71.5 | 91.3 | 80.2 | 70.6 | 77.1 |
| | CoCoOp | 26.6 | 95.0 | 58.9 | 68.2 | 70.4 | 84.3 | 85.3 | 70.8 | 93.5 | 77.1 | 70.6 | 72.8 |
| | PLOT | 41.0 | 95.0 | 62.4 | 83.2 | 80.6 | 95.4 | 83.9 | 73.3 | 92.9 | 82.0 | 70.4 | 78.2 |
| | MaPLe | 42.0 | 95.2 | 66.5 | 87.7 | 79.5 | 95.8 | 79.9 | 73.2 | 92.6 | 81.4 | 70.3 | 78.6 |
| | PromptSRC | **42.7** | 95.7 | 68.3 | 88.8 | **81.0** | **96.3** | 84.8 | **75.7** | 93.5 | 83.6 | 72.3 | 80.2 |
| | CutMix | 42.2 | 95.7 | 65.5 | 87.0 | 78.9 | 94.8 | 81.4 | 72.8 | 91.4 | 81.6 | 71.6 | 78.4 |
| | SelfMix | 31.5 | 95.5 | 64.4 | 76.8 | 70.8 | 91.9 | 83.6 | 72.8 | 93.0 | 79.5 | 71.4 | 75.6 |
| | Mixup | 38.8 | 95.5 | 63.6 | 89.1 | 78.0 | 94.0 | 82.0 | 73.1 | 93.0 | 81.9 | 71.3 | 78.2 |
| | CutOut | 37.2 | 95.3 | 63.9 | 88.4 | 75.7 | 93.6 | 82.5 | 73.3 | 91.7 | 81.2 | 71.4 | 77.7 |
| | **MVMP(Ours)** | 42.0 | **95.8** | **68.9** | **89.2** | 79.2 | 95.8 | **85.5** | 75.2 | **93.8** | **84.7** | **72.4** | **80.2** |
| **16 SHOTS** | CLIP | 45.4 | 95.4 | 70.0 | 87.2 | 80.4 | 97.4 | 82.9 | 73.3 | 85.3 | 82.1 | 67.3 | 78.8 |
| | CoOp | 43.4 | 95.6 | 69.9 | 84.9 | 83.1 | 97.1 | 84.2 | 74.7 | 91.9 | 82.2 | 71.9 | 79.9 |
| | CoCoOp | 31.2 | 95.2 | 63.0 | 73.3 | 71.6 | 87.8 | 87.3 | 72.2 | 93.3 | 78.1 | 70.8 | 74.9 |
| | PLOT | 46.7 | 95.8 | 71.0 | 91.8 | 83.0 | 97.0 | 86.0 | 76.0 | 93.2 | 83.9 | 71.3 | 81.4 |
| | MaPLe | 48.4 | 96.0 | 71.3 | 92.3 | 83.6 | 97.0 | 85.3 | 75.5 | 92.8 | 85.0 | 71.3 | 81.7 |
| | PromptSRC | 48.4 | 96.1 | 72.5 | 92.4 | **83.8** | 97.4 | 87.5 | 77.2 | 93.7 | **86.5** | 72.2 | 82.5 |
| | CutMix | **48.7** | 96.5 | 70.3 | 92.7 | 82.0 | 97.0 | 85.0 | 76.0 | 91.6 | 84.0 | 72.0 | 81.4 |
| | SelfMix | 41.5 | 96.3 | 69.1 | 80.4 | 73.7 | 95.2 | 86.5 | 74.6 | 93.1 | 81.2 | 72.0 | 78.5 |
| | Mixup | 46.7 | 96.4 | 69.0 | 91.4 | 80.9 | 95.8 | 86.0 | 75.6 | 93.4 | 83.7 | 72.7 | 81.1 |
| | CutOut | 44.5 | **96.4** | 67.0 | 92.1 | 79.5 | 96.0 | 86.2 | 75.5 | 93.2 | 82.5 | 72.6 | 80.5 |
| | **MVMP(Ours)** | 47.9 | 96.1 | **72.8** | **92.9** | 82.7 | **97.5** | **87.7** | **77.0** | **93.9** | 85.8 | **73.0** | **82.5** |

## A.4 ALGORITHM

In this section, we present the algorithm process. In Algorithm 1 and Algorithm 2, we show pseudo-code for the training and inference process of multi-vision multi-prompt for few-shot learning framework algorithm.

---

**Algorithm 1:** Training process of Multi-vision Multi-prompt for Few-shot Learning

---

**Input**: few-shot data $\mathcal{D} = \{\mathbf{x}, \mathbf{y}\}$, class label $y$, pre-trained CLIP model image encoder $f(\mathbf{x}, \theta_f, \boldsymbol{P_v})$, text encoder $g(\mathbf{y}, \theta_g, \boldsymbol{P_t})$, initial prompt $\boldsymbol{P} = (\boldsymbol{P_v}, \boldsymbol{P_t})$, high-layer prompt $\boldsymbol{P_h}$, fixed prompts $\boldsymbol{P}_{\text{fixed}}$, mixed area patch $\boldsymbol{I}$, Gaussian weight $w_i$, weight parameters $\lambda_1, \lambda_2, \lambda_3$, initial memory bank $M$, total epoch $E$.
**Output**: memory bank $M$, weighted high-layer prompt $\boldsymbol{P_h}$
**for** *all* $(\mathbf{x}_i, \mathbf{y}_i) \in \mathcal{D}$ **do**
    Randomly select an area $\boldsymbol{I}_1$ from $\mathbf{x}_i$, replaced the area $\boldsymbol{I}_2$ in $\mathbf{x}_j$ to generate new mixed sample $\mathbf{x}_{\text{mixed}}$
    Obtain original image and text feature $\boldsymbol{f_p}, \boldsymbol{g_p}$ with $\boldsymbol{f_p} \leftarrow f(\mathbf{x}_i, \theta_f, \boldsymbol{P_v})$,
    $\boldsymbol{g_p} \leftarrow g(\mathbf{y}_i, \theta_g, \boldsymbol{P_t})$
    Obtain mixed image feature $\boldsymbol{f}_{\text{mixed}}$ with $\boldsymbol{f}_{\text{mixed}} \leftarrow f(\mathbf{x}_{\text{mixed}}, \theta_f, \boldsymbol{P_v})$
    Obtain text feature diversity $\boldsymbol{g}_{\text{fixed}}$ with text encoder by $\boldsymbol{g}_{\text{fixed}} \leftarrow \frac{1}{D} g(\mathbf{y}_i, \theta_g, \boldsymbol{P}_{\text{fixed}})$
    **for** $e \in E$ **do**
        Calculate the self-augmentation loss $\mathcal{L}_{\text{vision}} = \text{MSE}(\mathcal{L}_{\text{ce}}(\boldsymbol{f}_{\text{mixed}}), \mathcal{L}_{\text{ce}}(\boldsymbol{f_p}))$
        Calculate the text loss $\mathcal{L}_{\text{text}} = L1 \text{ loss}(\boldsymbol{g_p}, \boldsymbol{g}_{\text{fixed}})$
        Calculate the cross-entropy CLIP loss $\mathcal{L}_{\text{ce}} = -\sum_i y_i \log(\text{sim}(\boldsymbol{f_p}, \boldsymbol{g_p})_i)$
        Total loss function $\mathcal{L} = \lambda_1 \mathcal{L}_{\text{vision}} + \lambda_2 \mathcal{L}_{\text{text}} + \lambda_3 \mathcal{L}_{\text{ce}}$
        Save prompt in Memory bank $M_{e+1} \leftarrow M_e \cup \{\boldsymbol{P_t} \cdot I(e)\}$
        Ensemble high-layer prompt with Gaussian weight $\boldsymbol{P_h} \leftarrow w_i \cdot \boldsymbol{P_h}$
    **end**
**end**

---

**Algorithm 2:** Inference process of Multi-vision Multi-prompt for Few-shot Learning

---

**Input**: Testing image data $\mathcal{D} = \{\mathbf{x}\}$, class label $y$, weighted high-layer prompt $\boldsymbol{P_h}$, fixed prompts $\boldsymbol{P}_{\text{fixed}}$, memory bank $M$.
**Output**: prediction of each image.
**for** *all* $(\mathbf{x}_i) \in \mathcal{D}$ **do**
    Sample $\boldsymbol{P}$ equally spaced prompts from $M$ with spacing $k$
    Let $\boldsymbol{P} = \{M[i], M[i+k], \ldots, M[i+(m-1)k]\}$
    Obtain text feature diversity $\boldsymbol{g}_{\text{fixed}}$ with $\{\boldsymbol{P}_{\text{fixed}}, \boldsymbol{P_h}\}$
    Obtain image feature $\boldsymbol{f}$
    **for** *each* $p \in \boldsymbol{P}$ **do**
        Obtain text feature $\boldsymbol{g_p}$ with $\{p, \boldsymbol{P_h}\}$
        Calculate the prediction distribution $l_p \leftarrow \arg\max(\boldsymbol{f} \cdot \boldsymbol{g_p})$
    **end**
    Calculate the fixed prediction distribution $l_{\text{fixed}} \leftarrow \arg\max(\boldsymbol{f} \cdot \boldsymbol{g}_{\text{fixed}})$
    Obtain the averaged prediction distribution for each image $i$,
    $l \leftarrow \frac{1}{m+1}\left(\sum_{p \in \boldsymbol{P}} l_{p_i} + l_{\text{fixed}_i}\right)$
**end**

---

## A.5 BASE-TO-NEW EXPERIMENT

In this section, we experiment with the generalization of multiple prompt methods in the base-to-new framework. In this experiment, we divide the 11 datasets into two parts based on CoCoOp (Zhou et al., 2022a) settings: base and new. Each part contains half of the classes. We trained a 16-shot model using only the base group and tested it on both the base and new groups. The aim is to evaluate

the generalization performance of the multi-prompt approach. The specific experimental results are presented in Table 6. CLIP itself has demonstrated superior generalization performance, while PLOT has shown progress for both the base and new groups. Additionally, MVMP has provided significant efficiency gains for both the base and new groups on most of the datasets, indicating that MVMP can maintain higher generalization performance in multi-prompt scenarios.

Table 6: Comparison of CLIP, PLOT and MVMP in the base-to-new generalization setting. The results justify the strong generalization of MVMP.

(a) Average

|  | Base | Unseen | Avg |
|---|---|---|---|
| CLIP | 69.4 | 74.2 | 71.8 |
| PLOT | 82.8 | 73.9 | 78.3 |
| MVMP | **84.1** | **74.8** | **79.5** |

(b) Caltech101

|  | Base | Unseen | Avg |
|---|---|---|---|
| CLIP | 96.8 | **93.9** | 95.4 |
| PLOT | 98.7 | 93.3 | 96.0 |
| MVMP | **98.7** | 93.8 | **96.3** |

(c) UCF101

|  | Base | Unseen | Avg |
|---|---|---|---|
| CLIP | 70.5 | 77.5 | 74.0 |
| PLOT | 85.7 | 78.8 | 82.3 |
| MVMP | **87.1** | **79.3** | **83.2** |

(d) ImageNet

|  | Base | Unseen | Avg |
|---|---|---|---|
| CLIP | 72.5 | 68.1 | 70.3 |
| PLOT | 76.1 | 68.1 | 72.1 |
| MVMP | **76.1** | **68.5** | **72.3** |

(e) OxfordPets

|  | Base | Unseen | Avg |
|---|---|---|---|
| CLIP | 91.3 | **97.0** | 94.2 |
| PLOT | 94.7 | 94.5 | 94.6 |
| MVMP | **95.8** | 96.3 | **96.1** |

(f) DTD

|  | Base | Unseen | Avg |
|---|---|---|---|
| CLIP | 53.2 | 60.1 | 56.7 |
| PLOT | 80.8 | 58.6 | 69.7 |
| MVMP | **83.1** | **61.0** | **72.1** |

(g) EuroSAT

|  | Base | Unseen | Avg |
|---|---|---|---|
| CLIP | 56.5 | 63.9 | 60.2 |
| PLOT | 91.9 | 64.1 | 78.0 |
| MVMP | **95.4** | **65.0** | **80.2** |

(h) Food101

|  | Base | Unseen | Avg |
|---|---|---|---|
| CLIP | 90.1 | 91.3 | 90.7 |
| PLOT | **90.8** | 91.3 | **91.1** |
| MVMP | 90.5 | **91.3** | 90.9 |

(i) StanfordCars

|  | Base | Unseen | Avg |
|---|---|---|---|
| CLIP | 63.4 | 74.8 | 69.1 |
| PLOT | 75.0 | 73.5 | 74.3 |
| MVMP | **77.3** | **75.5** | **76.4** |

(j) FGVCAircraft

|  | Base | Unseen | Avg |
|---|---|---|---|
| CLIP | 27.3 | 36.3 | 31.8 |
| PLOT | 41.5 | 36.2 | 38.9 |
| MVMP | **41.7** | **37.4** | **39.6** |

(k) SUN397

|  | Base | Unseen | Avg |
|---|---|---|---|
| CLIP | 69.3 | 75.4 | 72.4 |
| PLOT | 81.0 | 77.0 | 79.0 |
| MVMP | **82.3** | **78.8** | **80.6** |

(l) Flowers102

|  | Base | Unseen | Avg |
|---|---|---|---|
| CLIP | 72.2 | **77.9** | 75.1 |
| PLOT | 93.5 | 74.5 | 84.0 |
| MVMP | **97.5** | 76.5 | **87.0** |

Table 7: Detailed descriptions of the 11 experimental datasets, including the total number of training samples, the total number of test samples, and detailed content descriptions.

| Dataset | Classes | Test Size | Type |
|---|---|---|---|
| ImageNet (Deng et al., 2009) | 1,000 | 50,500 | Various objects, animals, scenes, etc. |
| OxfordPets (Parkhi et al., 2012) | 37 | 3,669 | Domestic pets, mainly cats and dogs |
| StanfordCars (Krause et al., 2013) | 196 | 8,041 | Automobiles, including different makes and models |
| Caltech101 (Fei-Fei et al., 2007) | 101 | 2,465 | Objects and scenes, including faces, watches, and animals |
| DTD (Cimpoi et al., 2014) | 47 | 1,692 | Textural patterns like striped, dotted, etc. |
| EuroSAT (Helber et al., 2019) | 10 | 8,100 | Satellite images of land cover like forests, roads, and fields |
| FGVCAircraft (Maji et al., 2013) | 100 | 3,334 | Various types of aircraft, including jets, propellers, etc. |
| Flowers102 (Nilsback & Zisserman, 2008) | 102 | 2,463 | Specific species of flowers like daisies, roses, etc. |
| Food101 (Bossard et al., 2014) | 101 | 30,300 | Different kinds of food dishes, including desserts and main courses |
| SUN397 (Xiao et al., 2010) | 397 | 19,850 | Various natural and man-made scenes, including forests, cities, and rooms |
| UCF101 (Soomro et al., 2012) | 101 | 3,783 | Videos of human actions, including sports, playing instruments, etc. |

## A.6 Additional Implementation Details

In this section, we provide a detailed description of the datasets and experiments. Table 7 shows the detailed description of datasets. We use the same prompt settings as CoOp (Zhou et al., 2022b), embedding the prompt at the "end" position of the text. Each benchmark is trained for 50 epochs with a batch size of 16. Initial text prompts are randomly generated with a length of 4 characters.

We sample from 50 prompts using 5 prompts for prediction. We use prompts with $V = T = 4VL$ in the first four transformer layers. Each of these prompts is randomly generated following a normal distribution. We use Gaussian weights with a standard deviation and mean both set to 30. We use a learning rate of 0.002 and set weights of the loss function, where $\lambda_1 = 2$, $\lambda_2 = 15$, $\lambda_3 = 5$. For the memory bank parameters, we use training accuracy as $\epsilon_e$ and the threshold $\beta = 50$. Additionally, we utilize 10 pre-defined manual prompts for distilling frozen CLIP text features (Radford et al., 2021). These texts are the seven with the highest training ImageNet accuracy in CLIP prompt and three randomly selected ones. The specific texts are as follows:

```
"itap of a {}."                    "art of the {}."
"a bad photo of the {}."           "a photo of the small {}."
"a origami {}."                    "a photo of a {}."
"a photo of the large {}."         "a bad photo of a {}."
"a {} in a video game."            "a photo of many {}."
```

