# OpenReview forum: "Multi-Vision Multi-Prompt for Few-Shot Learning in Vision-Language Model"
_ICLR.cc/2024/Conference — Submitted to ICLR 2024_

### Official Review · Reviewer_RSHG · 2023-10-29

**Soundness:** 3 good
**Presentation:** 2 fair
**Contribution:** 2 fair
**Rating:** 3
**Confidence:** 3

**Summary:**

This paper introduces the Multi-Vision Multi-Prompt (MVMP) method for few-shot learning in vision-language models such as CLIP. MVMP employs prompts of varying detail levels throughout training, utilizes a mixed self-augmentation framework to boost image diversity, and applies text distillation using multiple text features to enhance image-text alignment. Experimental results indicate that MVMP offers notable improvements compared to existing methods in few-shot environments.

**Strengths:**

The method consistently delivers great improvements across different tests, supported by comprehensive experimentation that showcases its effectiveness.

The enhancements achieved by the approach are multi-dimensional. By adjusting text prompts at distinct training stages, it is designed to gradually learn and adapt to a myriad of features. This systematic progression ensures that the model captures both broad and nuanced details effectively.

Beyond text prompt adjustments, the method places a significant emphasis on diversifying the dataset. This is achieved by strategies aimed at enriching both image and text data, ensuring a balanced and diverse training environment that contributes to the model's performance.

**Weaknesses:**

There's ambiguity regarding the benchmarking of the proposed methods. It's essential to ensure that each method has been equitably compared to provide a clear understanding of its efficacy.

Figure 6 indicates that a score of 73.8 is achieved by integrating multi-prompt, multi-vision, and text diversity. However, Table 2 seems to compare this with CLIP combined with other mix-up methods along, which would be more appropriate to compare it with just CLIP + multi-vision. According to Figure 6, multi-vision alone offers a mere 0.5% improvement. Given the renowned effectiveness of mix-up methods as data augmentation, it's not evident if the inclusion of multi-vision is truly indispensable. Moreover, the approach bears notable resemblance to mix-up, with distinctions primarily in implementation specifics.

Regarding multi-prompt, it might be more insightful if it was contrasted with a baseline where diverse prompts are introduced simultaneously across training stages. Introducing different prompts at separate stages complicates the algorithm, necessitating elements like a prompt bank. At present, it's challenging to discern the imperative of such staged or "curriculum" learning.

The paper's primary arguments seem to combine various methods from distinct directions. However, there's a lack of explicit justification underscoring the necessity and efficiency of each technique, particularly when benchmarked against its direct counterparts. Comparing with prior state-of-the-art methods doesn't necessarily bolster the paper's credibility, especially given the significant differences in implementation and methodologies.

**Questions:**

For Table 6, can we show CLIP +Multi-Vision; CLIP + Text Diversity additionally?
And then, for CLIP + Multi prompt; CLIP +Multi-Vision; CLIP + Text Diversity, they should be individually benchmarked with their own baselines.

---

### Official Review · Reviewer_bsQr · 2023-10-29

**Soundness:** 1 poor
**Presentation:** 1 poor
**Contribution:** 1 poor
**Rating:** 1
**Confidence:** 5

**Summary:**

This paper delves into the challenges of few-shot learning in vision-language models, particularly the Contrastive Language-Image Pre-Training model (CLIP). Recognizing that previous prompt-learning techniques typically depend on a singular prompt, which may not adequately differentiate between categories with diverse features and contexts, the authors introduce a novel method named Multi-Vision Multi-Prompt (MVMP). Designed specifically for CLIP in few-shot scenarios, MVMP avoids increasing model parameters by employing multiple prompts at various training phases and averages the outcomes. The study also introduces a combined self-augmentation framework and text distillation to boost the model's efficacy. Through rigorous experimentation, it's demonstrated that the MVMP method notably surpasses existing standards in few-shot learning classification tasks, enhancing accuracy rates by 4.6% and 2%.

**Strengths:**

1. An intuitive perspective to improve the inference capabilities of CLIP-like vision-language models.
2. Good empirical results. The new method demonstrates superior performance than its baselines in this paper.

**Weaknesses:**

Overall, the contribution of this paper is quite limited. The two basic components, i.e., multi-view augmentation and multiple prompts, of the newly introduced method have actually been explicitly explored in the community, yet this paper does not yield significant improvements with its simple and straightforward design. The weaknesses are detailed below:

1. The design of feeding multiple prompts into text encoder has long been a conventionally accepted method for zero-shot vision-language inference. For example, the original CLIP paper leverages 80 different text prompts for ImageNet inference. This technique is generally termed as prompt engineering but this paper lacks related discussions about that.
2. The mechanism of introducing text prompts for multiple transformer layers is also confusing. I understand multi-layer prompting might facilitate extracting multi-scale visual features for the image encoder, but why also applying it to text encoders? The paper does not give an explanation or discussion of this design, and it seems that the performance gain is merely because of introducing additional parameters.
3. Some important details are also missing. How are the prompt tokens processed in the transformers? Are prompts introduced in the i-th layer simply pruned after this layer, or are they retained until the last layer? These two protocols do make difference.
4. The Mixed Self-Augmentation looks more like an additional trick to enhance performance. This augmentation does not exhibit too much relation to your prompting framework, i.e., even without this augmentation, your method can achieve comparable performance, and when equipped with it, your baseline models also seem to be able to obtain the same level of improvements.
5. Some baseline performances might be underestimated. For example, why CLIP's test accuracy on ImageNet is only 32.1%? In CLIP's original paper it obtains 63.2% and 68.6% zero-shot accuracy with ViT-Base/32 and ViT-Base/16, respectively.
6. Comparisons are unfair. You method has both data augmentation and prompt learning but every baseline only has one of them. You should at least give ablation results of your method without augmentation/prompting or baselines with both sophisticated augmentation and prompting.
7. Minor comments: the paper is not well-written and hard to understand; many grammatical errors and imprecise expressions in the paper; In figure 6, the table should not be screenshotted to an image, but there are many ways to place images and tables side by side in Latex.

**Questions:**

See "Weaknesses".

---

### Official Review · Reviewer_gxiR · 2023-11-04

**Soundness:** 1 poor
**Presentation:** 2 fair
**Contribution:** 1 poor
**Rating:** 3
**Confidence:** 4

**Summary:**

This paper proposes MVMP, a multi-prompt method for CLIP in few-shot learning. The authors propose an image augmentation technique to diversify the visual samples, a text distillation technique to enhance textual features, and lastly, a Gaussian weight averaging for aggregating multilayer prompts learned from different network levels. Experiments show MVMP performs well on the 1-shot adaptation setting.

**Strengths:**

- This paper presents an easy-to-understand method that is especially effective on fewer-shot CLIP-based adaptation tasks.
- The experimental performance on the 1-shot setting is good and consistent.

**Weaknesses:**

- The paper writing and structuring can be substantially improved.
- All argument referencing Figure 2 is weak as they are not a direct reflection of the proposed method in this paper.
- Figure 4 is confusing, readers are unable to tell frozen parameters versus learnable parameters.
- The proposed image augmentation, i.e. mixup regions of image cross samples, is not well motivated and does not make much sense. The ablation result in Figure 6 also does not justify the necessity.
- Gaussian weighting for high-layer prompts seems like a fixed heuristic with no technical novelty.
- The results for cross-dataset benchmark evaluation are underwhelming, with the average accuracy higher than MaPLe by merely 0.2%.

**Questions:**

Please refer to the Weaknesses.

---

### Official Review · Reviewer_ihSe · 2023-11-04

**Soundness:** 2 fair
**Presentation:** 1 poor
**Contribution:** 3 good
**Rating:** 3
**Confidence:** 5

**Summary:**

This paper proposes a new method called Multi-Vision Multi-Prompt (MVMP) for few-shot learning in vision-language models. The authors claim that MVMP is the first approach to employ multiple prompts and a mixed self-augmentation framework to enhance the performance of CLIP in few-shot learning. The experimental validation shows that MVMP significantly outperforms the state-of-the-art methods in few-shot learning classification tasks.

**Strengths:**

+ The paper addresses an important problem in few-shot learning, which is the limited availability of labeled data. The proposed method, MVMP, is a novel approach that employs multiple prompts and a mixed self-augmentation framework to enhance the performance of CLIP in few-shot learning.

**Weaknesses:**

- There remain many unclear points regarding the proposed method.
  - What do $P_{fixed}$ and $P_t$ refer to? In other words, where do you obtain $P_{fixed}$ and $P_t$? Are $P_t$ and $P_{w}^{(e)}$ the same in the context? These notations are very confusing without clear explanations.
  - What does evaluation parameter $\epsilon_e$ mean? What is the formula to calculate it?
  - What does $H(x)$ mean in equation (8)? I assume you are referring to image features; however, it is somewhat challenging to interpret this notation if it's being introduced for the first time.
  - Is there any explanation why enforcing the consistency loss between original images and mixed images benefits few-shot learning?
  - Similarly for the text distillation loss? Please do not repeat the sentences mentioned in your paper in your rebuttal, since they do not provide any insights towards understanding your method.

- The conclusion in Figure 5 is that leveraging 5 prompts works the best. Here comes a question: how do you select the 5 prompts given so many combinations among hand-designed prompts?

- The paper claims that their approach doesn't introduce additional parameters or escalate computational costs. The question that arises is whether employing more prompts results in an increase in computational expenses. Nevertheless, the authors do not provide any comparison of computational costs between their method and other approaches.

- There are some other state-of-the-art few-shot approaches [1] missing for comparison, where the results are much better.

- Poor writing.
  - While the performance of few-shot learning can improve through meta-learning -> While the performance of few-shot learning can be improved through meta-learning
  - While the performance of few-shot learning can improve through meta-learning or image augmentation strategies, these approaches may increase computational cost and affect accuracy. This sentence is self-contradictory. You mentioned that the few-shot performance could be improved through those strategies, however, you later state that they affect accuracy.
  - improving accuracy by 4.6% and 2%. What do you mean?
  - our method improves overall accuracy by 2% to 4.6% -> our approach improves the overall accuracy by a range of 2% to 4.6%.


[1] Exploiting Category Names for Few-Shot Classification with Vision-Language Models

**Questions:**

See above.

---

### Meta-Review · Area_Chair_KXg7 · 2023-12-11

**Metareview:**

All reviewers are quite negative on this paper. They have raised a series of concerns. Many parts in the work are not clearly presented. Some important related and recent works are missing. The experimental comparisons are also not fair. No rebuttal is provided by the authors. Thus, the AC recommends rejection for this paper.

**Justification For Why Not Higher Score:**

There are many weaknesses in this paper, in terms of clarity, comparisons and experiments.

**Justification For Why Not Lower Score:**

N/A

---

### Decision · Program_Chairs · 2024-01-16

Reject